# One-step Noisy Label Mitigation

## Abstract

Mitigating the detrimental effects of noisy labels on the training process has become increasingly critical, as obtaining entirely clean or human-annotated samples for large-scale pre-training tasks is often impractical. Nonetheless, existing noise mitigation methods often encounter limitations in practical applications due to their task-specific design, model dependency, and significant computational overhead. In this work, we exploit the properties of high-dimensional orthogonality to identify a robust and effective boundary in cone space for separating clean and noisy samples. Building on this, we propose One-step Anti-Noise (OSA), a model-agnostic noisy label mitigation paradigm that employs an estimator model and a scoring function to assess the noise level of input pairs through just one-step inference, a cost-efficient process. We empirically demonstrate the superiority of OSA, highlighting its enhanced training robustness, improved task transferability, ease of deployment, and reduced computational costs across various benchmarks, models, and tasks. Our code is released at https://anonymous.4open.science/r/CLIP_OSN-E86C.

## 1 Introduction

Noise mitigation aims to handle the detriment of noisy labels encountered during the training process. The advancement of large-scale pre-training has significantly increased data scale to the trillion level. Much of this data is sourced from the internet, inevitably introducing considerable noise, which severely impedes the training process. This poses a substantial challenge for robust model training in various tasks, such as cross-modal matching (Huang et al., 2021; Zhang et al., 2024), image-classification (Sun et al., 2021; Yu et al., 2019), and image-retrieval (Liu et al., 2021).

Traditional noise mitigation approaches encounter several limitations that constrain their practical applicability: 1) **Task specificity:** Existing methods (Huang et al., 2021; Sun et al., 2021; Ibrahimi et al., 2022a) are tailored to specific tasks, limiting their applicability across different tasks. 2) **Model dependency:** Most noise mitigation techniques (Liu et al., 2021; Yang et al., 2023a) are tightly coupled with specific models, requiring extensive modifications for adaptation to different models. 3) **Computational cost:** Numerous existing methods necessitate dual-model collaborations (Huang et al., 2021; Yu et al., 2019) or multiple training passes (Huang et al., 2021), *i.e.*, they require at least two backward passes per training step, effectively doubling the computational expense and substantially increasing the training burden (see Figure. 1a).

To tackle these challenges, we use an external estimator to assess the noise level of each sample, ensuring a model-agnostic approach. This estimator adjusts the training loss by reducing the influence of noisy samples, driving their weights toward zero. Furthermore, multimodal pre-trained models have demonstrated remarkable task transferability due to their strong semantic capabilities. For instance, CLIP (Radford et al., 2021) unifies the paradigms of image-text retrieval and image classification through a shared embedding space (see Figure. 1b). It converts category labels into sentences, maps them into the shared embedding space, and then calculates the cosine similarity with the image representation to perform image classification. Inspired by this, we leverage multimodal pre-trained models as estimators and apply the shared embedding space to enable task transfer. In this case, only one additional inference process is required for each sample, significantly reducing the computational overhead compared to performing an extra backward pass.

Nonetheless, this paradigm introduces a new challenge: how to accurately identify noise based solely on cosine similarity scores generated by estimators. An ideal solution is to find a decision

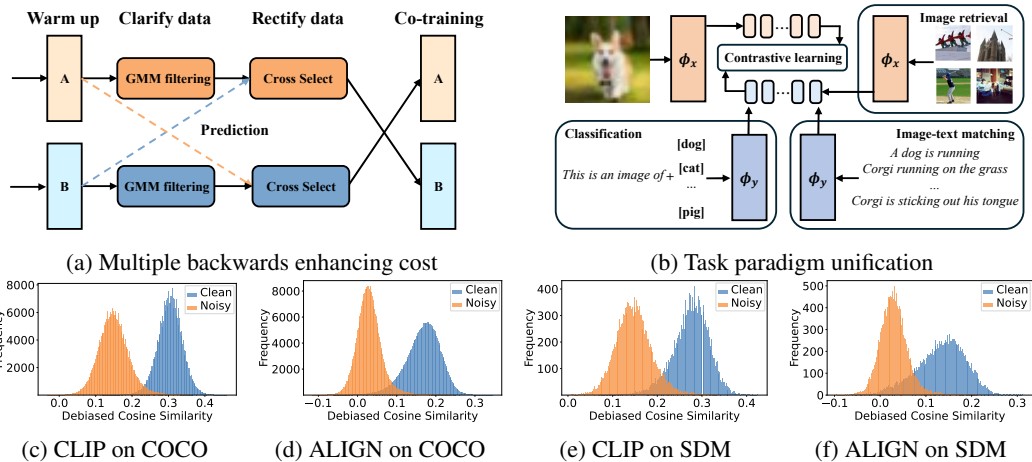

(a) Multiple backwards enhancing cost · (b) Task paradigm unification

(c) CLIP on COCO · (d) ALIGN on COCO · (e) CLIP on SDM · (f) ALIGN on SDM

Figure 1: **(a)** The current anti-noise paradigm with multiple backward significantly enhances the training overhead. **(b)** CLIP unifies the framework of image-text matching and image classification through a shared space. **(c-f)** Cosine similarity distribution of noise and clean data with 50% noise.

boundary that separates clean samples from noise and accurately handles overlapping samples near the boundary. Existing methods (Huang et al., 2021; Qin et al., 2022; Li et al., 2020; Zhang et al., 2024) typically attempt to build this boundary within the loss space, an isotropic space with uniform distribution, which creates only a narrow gap between noisy and clean samples. Moreover, the coarse handling of overlaps by integrating multi-model predictions often results in an unstable decision boundary. In contrast, the shared embedding space of pre-trained models is a high-dimensional, anisotropic space with an imbalanced distribution. Thus, a consideration is whether the properties of imbalanced anisotropic space can help to identify a more precise and robust decision boundary.

In this work, we delve into the decision boundary of pre-trained models employed as estimators to accurately differentiate between clean and noisy samples. We first investigate the cosine similarity distributions of clean and noisy samples, calculated using the multimodal pre-trained models CLIP (Radford et al., 2021) and ALIGN (Jia et al., 2021), on two datasets with a 50% noise ratio: MS-COCO (Lin et al., 2014) and SDM, as shown in Figure. 1c-1f. SDM is a dataset with the images generated by the Stable Diffusion Model (SDM) (Rombach et al., 2022) in some uncommon styles (see illustrations in Figure. 4). It is designed to explore how well-pre-trained models can distinguish unfamiliar domains that they rarely encounter during training. There are two interesting observations in Figure. 1c-1f: **(1)** The clean and noisy distributions of the same model on different datasets have a similar intersection point, suggesting the existence of a natural and stable boundary in distinguishing between clean and noisy samples. **(2)** The overlaps between the clean distribution and noisy distribution are minimal even in the unfamiliar domain dataset, indicating this boundary has strong potential for distinguishing between clean and noisy samples.

Building upon these two observations, we conduct an in-depth investigation and make the following contributions:

1. We figure out the origin of the intersection, attributing it to the shift in the orthogonal boundary induced by the cone effect. Furthermore, we provide a theoretical framework that proves and elaborates the stability and precision of this boundary in separating noisy and clean samples.

2. We provide a detailed explanation of the reliability of pre-trained models in general noise recognition, even in unfamiliar domains, grounded in the analysis of the pre-training process.

3. Build on this, we introduce One-Step Anti-Noise (OSA), a general model-agnostic paradigm for noise recognition that requires only one-step inference. Specifically, we utilize a pre-trained model as the estimator to maintain a shared embedding space. A scoring function, designed based on the properties of high-dimensional orthogonality, is then used to accurately handle overlaps by directly assigning a learning weight to each sample's loss according to its cosine similarity.

4. We conduct comprehensive experiments across a variety of challenging benchmarks, models, and tasks, demonstrating the effectiveness, generalization capabilities, and efficiency of our method.

## 2 BOUNDARY PRINCIPLE ANALYSIS

In Figure. 1c-1f, we observe a natural boundary emerging in the pre-trained model's ability to distinguish between clean and noisy samples. In this section, we explain the principle of boundary forming from high-dimensional perspectives, and how robust it is in general noise mitigation.

### 2.1 HYPOTHESIS: INTERACTION BOUNDARY IS SHIFTED FROM ORTHOGONAL BOUNDARY

We first elaborate on the gap extent between the positive and negative sides kept by the orthogonal boundary. Then, we present the reasoning behind the hypothesis that the intersection boundary in Figure. 1 is a shifted orthogonal boundary in the cone space.

**The orthogonal boundary largely separates the positive and negative sides.** High-dimensional orthogonality is a general phenomenon caused by dimension disaster, where the angles between randomly selected vectors typically approximate 90 degrees, suggesting the cosine similarity that trends toward zero. For instance, in a 1024-dimensional space, the probability of two random vectors having a cosine similarity within $[-0.1, 0.1]$ is approximately 99.86% (details are provided in Appendix. C.1). In this case, a natural boundary of cosine similarity zero forms, capably separating the positive side and negative side with a huge gap.

**Cone effect may induce orthogonal boundary shift.** Recent literature (Liang et al., 2022a; Bogolin et al., 2022; Ethayarajh, 2019) has demonstrated that the cone effect is a general phenomenon in deep neural networks, where the learned embedding subspace forms a narrow cone and the orthogonal boundary encounters a positive shift. Based on this, a hypothesis is that the interaction boundary in Figure. 1 is the shifted orthogonal boundary. To prove this, we simulate the process of selecting random vectors in high-dimensional space and randomly generate thousands of pairs mapped into the shared embedding space. We find that all similarity of these random vector pairs tends to a fixed

Table 1: The mean and variance of cosine similarity between randomly generated pairs.

| Model | Mean | Var |
|-------|------|-----|
| CLIP | 0.215 | 0.024 |
| ALIGN | 0.087 | 6e-4 |

value, with the low-variance cosine similarity almost lying in the middle of clean and noise distributions (see Table. 1). An interesting phenomenon is that if we compare the mean with the intersection points in Figure. 1c-1f, we find they are almost exactly the same. This suggests that the interaction boundary is highly likely to be a shifted orthogonal boundary in cone space.

### 2.2 THEORETICAL VERIFICATION OF THE INTERACTION BOUNDARY ORIGIN

Here, we theoretically investigate whether the origin of the interaction boundary is a shifted orthogonal boundary. We first show that (i) contrastive learning separates clean and noisy samples on opposite sides of the orthogonal boundary and (ii) The relative relationships of pairs' cosine similarity stays unchanged after transmitting into the narrow cone space. Based on (i) and (ii), we can confirm that the intersection boundary at the center of the clean and noisy distributions is the shifted orthogonal boundary.

**Contrastive learning empowers the separation of clean and noisy samples.** For an initialized model intending to learn an embedding space, both clean and noisy samples are treated as orthogonal random vectors since lacking semantic perception ability in the initial space. During contrastive training process, given $N$ sample pairs $\{(x_i, y_i)\}_{i=1}^N$, the embedding space is optimized through the cross-entropy loss (Eq. 1).

$$\mathcal{L}_{ce} = \frac{1}{N} \sum_{i=1}^{N} \log \frac{\exp(m_{ii})}{\sum_{j=1}^{N} \exp(m_{ij})}, \tag{1}$$

where $M \in \mathbb{R}^{N \times N}$ represents the cosine similarity matrix of $N$ sample pairs during training process. Each element $m_{ij} \in M$ denote the cosine similarity between $x_i$ and $y_j$. The diagonal elements $m_{ii}$ denote the cosine similarities of positive pairs, while the non-diagonal elements $m_{ij}$ represent the cosine similarities of negative pairs.

To minimize $\mathcal{L}_{ce}$ during training, two subprocesses occur: the diagonal elements of the matrix (*i.e.*, clean pairs) are optimized to the positive side of the orthogonal boundary, while the non-diagonal elements (equivalent to noise pairs) are optimized to the negative side. Consequently, the distributions of these two types of samples are on opposite sides of the orthogonal boundary.

**Relative relationship unchanged in transmitting process.**   We study how the boundary shifts from the entire space to the narrow cone in the neural network. The following theorem shows that the cosine similarity will be proportionally scaled to the target narrow cone, while still maintaining a boundary with properties similar to the orthogonal boundary. In other words, vectors with cosine similarity smaller than the orthogonal boundary in the original space remain smaller than the shifted boundary in the narrow cone space, while those larger remain larger.

**Theorem 1** (Proportional shift of boundary). *Let $\mathbb{R}^{d_{in}}$ be the original space before being transmitted in a neural network. Suppose $u, v \in \mathbb{R}^{d_{in}}$ are any two random vectors with $\cos(u, v) \approx 0$. $u_c, v_c \in \mathbb{R}^{d_{in}}$ is a pair of clean vectors with $\cos(u_c, v_c) > 0$, while $u_n, v_n \in \mathbb{R}^{d_{in}}$ is a noisy pair with $\cos(u_n, v_n) < 0$. Given a Neural Network $F(x) = f_t(f_{t-1}(\ldots f_2(f_1(x)))) \in \mathbb{R}^{d_{out}}$ with $t$ layers. $f_i(x) = \sigma_i(\mathbf{W}_i x + \mathbf{b}_i)$ denotes $i^{th}$ layer, where $\sigma(\cdot)$ indicates activation function. $\mathbf{W}_i \in \mathbb{R}^{d_{out}^i \times d_{in}^i}$ is a random weight matrix where each element $\mathbf{W}_i^{k,l} \sim \mathcal{N}(0, 1/d_{out}^i)$ for $k \in \left[d_{out}^i\right]$, $l \in \left[d_{in}^i\right]$, and $\mathbf{b}_i \in \mathbb{R}^{d_{out}^i}$ is a random bias vector such that $\mathbf{b}_i^k \sim \mathcal{N}(0, 1/d_{out}^i)$ for $k \in \left[d_{out}^i\right]$. Then, there always be a boundary $\beta$, satisfying:*

$$\cos(F(u_n), F(v_n)) < \cos(F(u), F(v)) \approx \beta < \cos(F(u_c), F(v_c)). \tag{2}$$

Theorem. 1 shows that the relative relationship of pairs in the original entire space, will not change after transmitting to the narrow cone space of the trained model, and there is always a boundary $\beta$ concentrated on most random vectors. In Appendix. C.2, we provide a detailed statement and proof of the Theorem.

## 2.3 QUALITATIVE ANALYSIS OF ROBUSTNESS AND APPLICABILITY

Next, we perform a qualitative analysis to explore (i) the robustness and generality of the boundary in distinguishing between clean and noisy samples, and (ii) how the boundary's properties can be leveraged to achieve more reasonable and precise overlap handling.

**How about the boundary robustness even in unfamiliar domains?**   Although the boundary's ability to distinguish clean and noisy samples is proven, its robustness and generality still require further exploration. For practical pre-training, it must maintain accuracy and robustness even in unfamiliar domain datasets. Since the capabilities of the pre-trained model are difficult to quantify, we conduct a qualitative analysis from the perspective of pre-trained model inference. The models pre-trained on millions of samples already possess somewhat semantic understanding capabilities. Given a positive pair from an unseen domain, due to the contrastive learning process during pre-training, it still has a strong likelihood of moving toward the positive side of the boundary, while the negative pair tends toward the negative side. Although the cosine similarity difference might be slight, as we have shown in Section. 2.1, the boundary constructs a significant gap from the perspective of high-dimensional orthogonality.

**How to handle the overlaps through imbalanced probability?**   Due to the properties of orthogonal boundary, as cosine similarity decreases and approaches zero from the positive side, the probability of positive samples sharply decreases. Therefore, we can design a scoring function to annotate the cleanliness of samples. This function should satisfy two requirements: for samples with cosine similarity less than or equal to zero, which are almost certainly noise, the function should assign them a weight of zero. For samples with cosine similarity greater than zero, the function gradient should increase rapidly as the cosine similarity moves further from zero.

## 3 METHOD

In this section, we present our One-Step Anti-Noise (OSA) paradigm with a workflow shown in Figure. 2. We first define the pair-based noise mitigation tasks for image-text matching, image

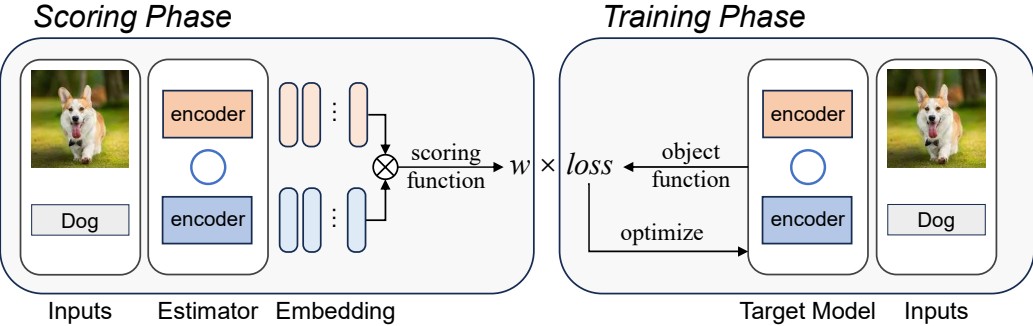

Figure 2: The workflow of OSA. In the anti-noise process, there are two phases: Scoring Phase and Training Phase. In the Scoring Phase, a pair is mapped to a shared embedding space by estimators. Then the cosine similarity is transformed to a weight $w$ by a scoring function. In the Training Phase, the weight $w$ is directly multiplied with the loss to instruct the optimization.

classification, and image retrieval tasks in Sec. 3.1. Consequently, the detailed description of OSA is clarified in Sec. 3.2.

## 3.1 TASK DEFINITION

Let $\mathcal{D} = \{(x_i, y_i, c_i)\}_{i=1}^N$ denote a paired dataset, where $(x_i, y_i)$ represents the $i$-th pair in the dataset, and $c_i$ indicates a noise label for that pair. Specifically, when $c_i = 0$, $(x_i, y_i)$ forms a correct (paired) match, while $c_i = 1$ denotes an incorrect (unpaired) match. The objective of noise mitigation in contrastive learning is to construct a shared embedding space that brings $x_i$ and $y_i$ closer when $c_i = 1$. In different tasks, $x_i$ and $y_i$ are distinct data types. For instance, in the image-text retrieval task, $x_i$ and $y_i$ represent images and texts, respectively. In the image classification task, $x_i$ and $y_i$ represent images and categories, respectively. In the image retrieval task, $x_i$ and $y_i$ represent images and relevant images, respectively. The paired sample $(x, y)$ could be encoded into a shared embedding space by corresponding encoders $\phi_x(\cdot)$ and $\phi_y(\cdot)$. Afterward, the cosine similarity $s(x, y)$ is calculated through Eq. 3 as semantic relevance of $(x, y)$ to guide the training.

$$s(x, y) = \frac{\phi_x(x)}{\|\phi_x(x)\|} \cdot \frac{\phi_y(y)}{\|\phi_y(y)\|}. \tag{3}$$

## 3.2 ONE-STEP ANTI-NOISE

The workflow of our noise mitigation approach OSA is depicted in Figure. 2. Initially, we utilize an estimator model to encode the input pair to a shared embedding space and continue to compute the cosine similarity between the paired embedding. Afterward, the cosine similarity is converted to a cleanliness score $w_i, (0 \le w_i \le 1)$ through a scoring function designed based on orthogonal properties (Section. 2.3). This score quantifies the clean degree of the sample, the smaller $w_i$ is, the noisier the sample.

During the target model training phase, this cleanliness score is used as a weight, directly multiplied by the loss of the corresponding sample to facilitate selective learning. This noise mitigation process, being solely dependent on the estimator model, is readily adaptable to the training of various target models by simply adding an extra coefficient to the loss function, ensuring the model-agnostic property. Therefore, the key of our noise mitigation approach revolves around the estimator model and noise score assessment.

### 3.2.1 ESTIMATOR MODEL

**Estimator model selection.** In our approach, the Estimator Model must satisfy two critical requirements: 1) effectively mapping input pairs into a unified embedding space and 2) possessing basic semantic understanding capabilities. To meet these requirements, we employ CLIP (Radford et al., 2021), a commonly used multimodal pre-trained models, as our estimator model. It is

equipped with a text encoder $\phi_t(\cdot)$ and an image encoder $\phi_v(\cdot)$, enabling it to perform basic zero-shot tasks efficiently.

**Domain adaptation (Optional).** While we have performed a qualitative analysis of the zero-shot pre-trained model's robustness on out-of-domain data in Section. 2.3, and shown strong robustness for edge cases in Figure. 1, considering the domain diversity in real-world scenarios, we provide an optional Domain Adaptation (DA) approach to enhance the estimator model's adaptability when encountering edge domains. Following NPC (Zhang et al., 2024), we first employ a Gaussian Mixture Model (GMM) coupled with strict selection thresholds to ensure the absolute cleanliness of the chosen samples. We afterward implement a warm-up phase with few steps, allowing the estimator model to better understand the semantics of the target domain. Notably, this trick is only optional for our methods. Through multiple experiments, we found that even without domain adaptation, the zero-shot CLIP model performs exceptionally well across various scenarios.

### 3.2.2 NOISE SCORE ASSESSMENT

**Spatial Debiasing.** The cone effect phenomenon has been demonstrated as a general phenomenon for deep neural networks, typically resulting in a narrow embedding space that causes a shift of space center to a narrow cone center (Liang et al., 2022a). Specifically, when paired randomly generated inputs are mapped into a shared embedding space through model encoders, the resultant vectors exhibit an average cosine similarity that deviates from zero and tends to another fixed angle. To counteract this shift and mitigate its impact on the estimator's ability to accurately recognize noises through high-dimensional orthogonality, a random sampling method is developed. We begin by constructing $K$ random sample pairs $\mathcal{R} = \{(x_j, y_j) \mid j = 1, 2, \dots, K\}$ and processing them through the estimator's encoder to generate a set of vectors. Then the average cosine similarity among these vectors will be calculated as the space shift $\beta$ through:

$$\beta = \frac{\sum_{j=1}^{K} s(x_j, y_j)}{K}.$$ (4)

**Scoring Function.** After spatial debiasing, we employ a scoring function $w(\cdot)$ to evaluate the cleanliness of the input pair $(x, y)$. In section. 2.3, we have elaborate how to handle overlaps based on the orthogonal boundary property. For an estimator model trained on millions of samples using contrastive learning, clean pairs (diagonal elements) are optimized to positive side, while noise pairs (non-diagonal elements) are optimized to negative side. Given unfamiliar pairs, the model also tends to map clean pairs towards positive and noisy pairs towards negative. Despite the potentially slight similarity difference between clean and noisy pairs, high-dimensional orthogonality ensures a substantial gap between them. In this case, a negative cosine similarity $s(x, y)$ computed by the estimator, indicating the pair is almost certainly noise, should be assigned a score of zero. For samples with $s(x, y)$ greater than zero, the probability of the sample being positive sharply decreases as the cosine similarity approaches zero from the positive side. Therefore, the function gradient should increase rapidly as the cosine similarity moves further from zero. To systematically score the noise, we design the scoring function as:

$$w(x, y, \beta) = \begin{cases} 0 & , s(x, y) - \beta \leq 0 \\ -(s(x, y) - \beta)^2 (s(x, y) - \beta - 1) & , otherwise \end{cases}$$ (5)

**Re-weight Training.** After scoring, the target model can selectively learn from the samples by re-weighting the loss. Noise samples with smaller weights will have a reduced impact on model updates and will be effectively mitigated. For a sample $(x, y)$, let $\mathcal{L}_{x,y}$ denote its loss, the re-computed loss $\mathcal{L}_{re}$ is defined as:

$$\mathcal{L}_{re} = w(x, y, \beta) \times \mathcal{L}_{x,y}.$$ (6)

## 4 EXPERIMENTS

In this section, we present experiments on multiple datasets with label noise, demonstrating the effectiveness of our methods. Firstly, we describe the datasets, metrics, and implementation details. Then, we report our results on several downstream tasks. Lastly, we conduct ablation studies to

show how each part of our method contributes and examine how these parts interact. The literature involved in our experiments and richer related work are detailed in Appendix. B.

## 4.1 EVALUATION SETTING

In this section, we briefly introduce the datasets and evaluation metrics used in the experiments. For more dataset and implementation details, please refer to Appendix. A.

**Datasets.** We evaluate our method on three downstream tasks with noisy labels, including one multimodal task and two visual tasks. For the cross-modal matching task, we perform experiments on the MSCOCO (Lin et al., 2014) and Flickr30K (Young et al., 2014) datasets. Following NPC (Zhang et al., 2024), we further carry out evaluations on a real-world noisy dataset CC120K. For image classification tasks, experiments are conducted under three subsets of WebFG-496 (Sun et al., 2021)—Aircraft, Bird, and Car. For image retrieval tasks, we conduct experiments on the CARS98N dataset under PRISM (Liu et al., 2021) setting.

**Evaluation Metrics.** For the image-text matching task, the recall value of the top-K retrieved results (R@K) is used. For classification tasks, accuracy serves as the evaluation metric. For the image retrieval task, we use Precision@1 and mAP@R for evaluation.

## 4.2 COMPARISONS WITH STATE OF THE ARTS

Table 2: Comparison on noisy MS-COCO.

| Noise ratio | Method | MS-COCO 1K | | | | | | MS-COCO 5K | | | | | |
| | | | i2t | | | t2i | | | i2t | | | t2i | |
| | | R@1 | R@5 | R@10 | R@1 | R@5 | R@10 | R@1 | R@5 | R@10 | R@1 | R@5 | R@10 |
|---|---|---|---|---|---|---|---|---|---|---|---|---|---|
| 0% | VSE∞ | 82.0 | **97.2** | 98.9 | 69.0 | 92.6 | 96.8 | 62.3 | 87.1 | **93.3** | 48.2 | **76.7** | 85.5 |
| | PCME++ | 81.6 | **97.2** | 99.0 | **69.2** | **92.8** | 97.1 | 62.1 | 86.8 | **93.3** | 48.1 | **76.7** | **85.5** |
| | PAU | 80.4 | 96.2 | 98.5 | 67.7 | 91.8 | 96.6 | 63.6 | 85.2 | 92.2 | 46.8 | 74.4 | 83.7 |
| | NPC | **82.2** | 96.5 | **98.7** | 68.3 | 92.0 | **98.7** | 65.4 | **87.3** | 93.1 | 48.5 | 75.4 | 84.4 |
| | CLIP | 80.1 | 95.7 | 98.2 | 67.1 | 91.4 | 96.6 | 62.9 | 84.9 | 91.6 | 46.5 | 73.8 | 82.9 |
| | **+OSA** | **82.2** | 96.5 | **98.7** | 68.8 | 92.1 | 96.7 | **65.6** | 86.8 | 92.9 | **49.1** | 76.2 | 84.8 |
| | ALIGN | 84.9 | 97.3 | **99.0** | 70.5 | 92.8 | 97.2 | 69.6 | **89.9** | 94.5 | 50.5 | 77.5 | 85.7 |
| | **+OSA** | **85.3** | **97.4** | **99.0** | **71.4** | **93.1** | **97.3** | **69.8** | **89.9** | **94.8** | **51.4** | **78.2** | **86.3** |
| 20% | VSE∞ | 78.4 | 94.3 | 97.0 | 65.5 | 89.3 | 94.1 | 58.6 | 83.4 | 89.9 | 45.0 | 72.9 | 81.7 |
| | PCME++ | 78.4 | 95.9 | 98.4 | 64.9 | 90.8 | 96.1 | 57.7 | 83.9 | 91.0 | 43.2 | 72.3 | 82.4 |
| | PAU | 78.2 | 95.2 | 98.1 | 64.5 | 90.0 | 95.4 | 59.3 | 82.9 | 90.4 | 44.2 | 71.3 | 81.3 |
| | NPC | 79.9 | 95.9 | 98.4 | 66.3 | 90.8 | **98.4** | 61.6 | 85.4 | 91.6 | 46.0 | 73.4 | 82.9 |
| | CLIP | 76.0 | 94.3 | 97.5 | 63.4 | 89.0 | 94.8 | 55.3 | 79.1 | 86.9 | 41.0 | 68.8 | 79.3 |
| | **+OSA** | 81.6 | 96.2 | 98.5 | 68.9 | 92.0 | 96.6 | 65.8 | 86.4 | 92.5 | 48.7 | 76.1 | 84.5 |
| | ALIGN | 79.4 | 95.7 | 98.2 | 66.2 | 90.8 | 96.1 | 60.9 | 84.5 | 91.0 | 46.3 | 73.6 | 82.3 |
| | **+OSA** | **85.1** | **97.4** | **99.1** | **70.9** | **93.0** | **97.3** | **69.7** | **90.0** | **94.7** | **50.9** | **77.8** | **86.2** |
| 50% | VSE∞ | 44.3 | 76.1 | 86.9 | 34.0 | 69.2 | 84.5 | 22.4 | 48.2 | 61.1 | 15.8 | 38.8 | 52.1 |
| | PCME++ | 74.8 | 94.3 | 97.7 | 60.4 | 88.7 | 95.0 | 52.5 | 79.6 | 88.4 | 38.6 | 68.0 | 79.0 |
| | PAU | 76.4 | 94.1 | 97.6 | 62.3 | 88.5 | 94.6 | 57.3 | 81.5 | 88.8 | 41.9 | 69.4 | 79.6 |
| | NPC | 78.2 | 94.4 | 97.7 | 63.1 | 89.0 | **97.7** | 59.9 | 82.9 | 89.7 | 43.0 | 70.2 | 80.0 |
| | CLIP | 73.9 | 93.0 | 97.2 | 60.1 | 87.3 | 94.0 | 54.1 | 78.5 | 86.6 | 39.7 | 67.2 | 77.5 |
| | **+OSA** | 80.4 | 96.2 | 98.6 | 67.8 | 91.6 | 96.4 | **64.0** | 85.5 | 91.9 | 47.9 | 74.6 | 83.8 |
| | ALIGN | 78.0 | 95.8 | 98.5 | 65.4 | 90.3 | 96.0 | 60.1 | 84.3 | 91.2 | 45.2 | 72.8 | 82.1 |
| | **+OSA** | **84.3** | **97.0** | **98.9** | **70.0** | **92.5** | 97.0 | **68.5** | **89.2** | **94.2** | **50.0** | **77.0** | **85.4** |

**Results on MSCOCO.** To fairly demonstrate the effectiveness of our method, we compare OSA with various robust learning image-text matching approaches using the same ViT-B/32 CLIP as backbone, including VSE∞ (Chen et al., 2021), PCME++ (Chun, 2023), PAU (Li et al., 2023), NPC (Zhang et al., 2024). Besides, we separately employ OSA on both CLIP (Radford et al., 2021) and ALIGN (Jia et al., 2021). The results in Table. 2 show that OSA outperforms all previous approaches on all metrics with a huge gap. In the more challenging MS-COCO 5K set with 50% noise ratio, OSA surpasses the SOTA method NPC in the R@1 for both image-to-text (i2t) and text-to-image (t2i) matching by 8.6% and 7.0%, respectively. Another phenomenon is that as the noise ratio increases from 0% to 50%, all other methods encounter severe performance drop, with

an averaging drop of 5.05% for NPC across four R@1 metrics. In contrast, OSA exhibits only a slight decrease of 1.275%, showcasing the accuracy and robustness of OSA in anti-noise tasks.

**Results on Flickr30K.** To further demonstrate the generalization ability of OSA, we evaluate on the Flickr30K dataset and compare with several anti-noise methods, including NCR (Huang et al., 2021), DECL (Qin et al., 2022), BiCro (Yang et al., 2023a), and NPC (Zhang et al., 2024). The results are presented in Table. 8 of Appendix. It is evident that OSA consistently outperforms all models on the R@1 metric. Notably, compared with the baseline CLIP, training with OSA at a 60% noise ratio achieves 20.9% R@1 improvement for i2t and a 22.3% R@1 improvement in t2i, further indicating the effectiveness of OSA on noise mitigation. Additionally, OSA demonstrates similar noise robustness on the Flickr30K dataset as observed on MSCOCO, with only 1.4% R@1 drop on i2t and 1.2% R@1 drop on t2i ranging from 0% noise to 60% noise, while all of the other anti-noise approaches hardly resist the detriment from high-ratio noise. All of these results demonstrate the effectiveness and robustness of OSA on anti-noise tasks.

**Results on CC120K.** To further verify the reliability of OSA in real scenarios, we conduct evaluations on a large-scale real-world noisy dataset, CC120K, with 3%-20% noise ratio. The results shown in Table. 3 indicate that OSA outperforms the current state-of-the-art method NPC, even in larger-scale real-world domains. This demonstrates the feasibility and generality of OSA even in practical training scenarios.

Table 3: Comparison on real-world noisy dataset CC120K.

| Method | i2t | | | t2i | | |
|---|---|---|---|---|---|---|
| | R@1 | R@5 | R@10 | R@1 | R@5 | R@10 |
| NPC | 71.1 | 92.0 | **96.2** | 73.0 | 90.5 | **94.8** |
| CLIP | 68.8 | 87.0 | 92.9 | 67.8 | 86.4 | 90.9 |
| +OSA | **73.1** | **92.2** | 95.7 | **73.9** | **91.2** | 94.7 |

Table 4: Results of other image-based tasks.

| Method | Image Classification | | | Image Retrieval | |
|---|---|---|---|---|---|
| | Aircraft Acc | Bird Acc | Car Acc | Prec. | mAP |
| Baseline | 65.44 | 62.29 | 75.90 | 71.69 | 18.16 |
| **+OSA** | **73.18** | **70.50** | **80.19** | **78.45** | **24.99** |

**Results on Other Downstream Tasks.** To validate the transferability of OSA across different tasks, we evaluate it on two additional tasks: image classification and image retrieval. The results are presented in Table. 4. The baseline method for both tasks leverages contrastive learning. In the image classification task, OSA outperforms the baseline by 7.74%, 8.21%, and 4.28% on the Aircraft, Bird, and Car subsets, respectively. In the image retrieval task, OSA improves performance by 6.76% in precision and 6.83% in mAP. These improvements demonstrate the strong task transferability and generality of OSA.

### 4.3 TARGET MODEL-AGNOSTIC ANALYSIS

OSA is an architecture-agnostic paradigm that can be easily adapted to various models. To verify its model-agnostic property, we evaluate it across models with different architectures. Subsequently, we apply it to other anti-noise models to demonstrate its generalization capability in noise mitigation.

**Architecture-agnostic Analysis.** The effectiveness of OSA on Vision Transformer (ViT) has been proven in Section. 4.2. We further explore the generality of OSA on target models with other architectures. Specifically, we deploy OSA above the VSE++ (Faghri et al., 2018) model with two different architecture types: ResNet-152 (He et al., 2016) and VGG-19 (Simonyan & Zisserman, 2014). These two architectures have revealed significant sensitivity and vulnerability to noise (Huang et al., 2021). In this experiment, all estimator models employ zero-shot CLIP and we utilize the original VSE++ as our baseline. The results in Table. 5 indicate a significant performance degradation emerged for the baseline methods in noisy setting, while a stable performance is achieved after employing OSA. The stable performance on these two noise-vulnerable architectures fully demonstrates that OSA possesses the architecture-agnostic property.

**Adaptability to Other Anti-Noise Models.** Theoretically, OSA can be adapted to any target model, providing noise resistance. However, can OSA further enhance the robustness of models

Table 5: The results of the target model with different architectures on noisy MSCOCO.

| Noise ratio | Method | Architecture | MS-COCO 1K | | | | | | MS-COCO 5K | | | | | |
|---|---|---|---|---|---|---|---|---|---|---|---|---|---|---|
| | | | i2t | | | t2i | | | i2t | | | t2i | | |
| | | | R@1 | R@5 | R@10 | R@1 | R@5 | R@10 | R@1 | R@5 | R@10 | R@1 | R@5 | R@10 |
| 0% | Baseline | ResNet-152 | 58.9 | **86.9** | **93.8** | 44.2 | 77.9 | **88.3** | 34.9 | **64.3** | **76.1** | 23.3 | **50.9** | **64.2** |
| | **+OSA** | | 58.9 | 86.2 | 93.7 | **44.3** | 77.9 | 87.9 | **35.0** | 64.1 | 76.0 | **23.5** | 50.8 | 63.9 |
| | Baseline | VGG-19 | 49.6 | 79.4 | 89.1 | 38.0 | 72.9 | **84.7** | **26.9** | 54.2 | 66.8 | 18.7 | 43.8 | 56.8 |
| | **+OSA** | | 50.1 | 80.0 | 89.3 | 38.3 | 73.0 | 84.6 | 26.6 | **54.4** | 67.4 | 18.8 | 43.9 | 57.3 |
| 20% | Baseline | ResNet-152 | 45.8 | 70.3 | 83.7 | 36.1 | 68.4 | 79.7 | 26.0 | 48.4 | 58.3 | 18.3 | 42.0 | 54.0 |
| | **+OSA** | | 58.1 | 86.1 | 93.2 | 43.4 | 76.8 | 87.2 | 33.7 | 62.6 | 74.5 | 22.5 | 49.7 | 62.8 |
| | Baseline | VGG-19 | 33.2 | 67.1 | 81.5 | 25.9 | 58.0 | 71.4 | 13.7 | 35.0 | 49.2 | 10.7 | 29.9 | 41.9 |
| | **+OSA** | | 49.3 | 79.1 | 88.6 | 37.2 | 71.9 | 83.8 | 25.2 | 53.3 | 65.3 | 17.9 | 42.6 | 55.9 |
| 50% | Baseline | ResNet-152 | 28.4 | 61.2 | 75.2 | 5.2 | 14.0 | 19.5 | 11.0 | 31.0 | 43.6 | 1.6 | 6.0 | 9.2 |
| | **+OSA** | | 55.0 | 84.0 | 92.0 | 40.7 | 74.7 | 85.6 | 30.8 | 60.2 | 72.3 | 20.9 | 46.6 | 60.0 |
| | Baseline | VGG-19 | 2.5 | 9.8 | 16.2 | 0.1 | 0.5 | 1.0 | 0.5 | 2.5 | 4.4 | 0.0 | 0.1 | 0.2 |
| | **+OSA** | | 47.1 | 77.7 | 87.6 | 35.7 | 70.3 | 82.8 | 24.0 | 51.5 | 64.0 | 16.9 | 40.8 | 54.2 |

specifically designed for noise mitigation? To investigate this, we applied OSA to the current state-of-the-art model, NPC (Zhang et al., 2024). As shown in Table. 9 of Appendix, even for noise-mitigating models, OSA consistently improves training robustness. This finding further demonstrates the broad adaptability of OSA across different model types.

## 4.4 ESTIMATOR MODEL ANALYSIS.

The estimator model is the basis of OSA's anti-noise capability. In this section, we explore the impact of different estimator models on noise mitigation, and examine the impact of domain adaptation in noise mitigation. In Table. 10 of Appendix, we investigate four types of estimators: "None" refers to training CLIP directly without using OSA. "CLIP (w/o DA)" and "ALIGN (w/o DA)" represent using CLIP and ALIGN without domain adaptation as estimators, respectively, *i.e.*, zero-shot CLIP and ALIGN. "CLIP (w DA)" indicates the CLIP with domain adaptation. The target models are all CLIP. We can observe that both of CLIP and ALIGN as estimators significantly enhance the target model performance stability when learning with noise, indicating that the choice of estimator is very flexible. Both CLIP and ALIGN demonstrate exceptional performance when served as estimators. The other phenomenon is that the zero-shot CLIP model shows comparable performance to the domain-adapted CLIP with a even better performance at lower noise ratios. This indicates that zero-shot CLIP, as an estimator, already performs exceptionally well in noise mitigation. The domain adaptation is unnecessary. This further enhances the deployment convenience of OSA.

## 4.5 NOISE ASSESSMENT ACCURACY

**Noise Detection Accuracy Analysis.** To figure out how accurate OSA is in recognizing noise, we evaluate the accuracy and recall on CLIP without Domain-Adaptation (w/o DA) and CLIP with Domain-Adaptation (w DA) on noisy MSCOCO. We utilize zero as the threshold to roughly divide pairs into noise and clean sets, respectively. Concretely, we classify scores less than or equal to 0 as noise, and scores greater than 0 as clean. The Accuracy means the proportion of the clean pairs correctly classified into the clean set, while the Recall indicates the noisy pairs correctly classified into the noisy set. The results presented in Table. 6 indicates the powerful noise recognizing capability of OSA. The remarkable performance on CLIP (w/o DA) fully demonstrates the generality of OSA. Another notable phenomenon is that all recall scores converge towards 100, indicating that OSA achieves greater accuracy in noise detection. This suggests that OSA can almost entirely eliminate the impact of noise on training.

**Noise Re-weighting Accuracy Comparison.** Some anti-noise methods, like NPC, also employ loss re-weighting for optimization. To assess whether our method assigns relatively smaller weights to noise than these methods, we first analyze the weights generated by NPC and OSA. Due to differences in weight scales across methods, a direct comparison is unfair. Therefore, to unify the scale, we adopt a ranking-based approach, sorting weights in descending order and calculating the Mean Noise Rank. This metric evaluates whether smaller weights are consistently assigned to noisy samples relative to clean ones. Our experiments use 2,000 randomly selected samples from the

MSCOCO dataset under two noise conditions: 20% noise (370 noisy samples) and 50% noise (953 noisy samples). The theoretical optimal Mean Noise Ranks, where all noisy weights are ranked last, are 1815.5 and 1524.0, respectively. Results presented in Table. 11 of Appendix show that OSA achieves a higher Mean Noise Rank compared to NPC, demonstrating greater accuracy in re-weighting. Moreover, OSA's rankings are nearly optimal (20% noise: 1809.1 for OSA versus 1815.5 optimal; 50% noise: 1520.7 for OSA versus 1524.0 optimal). This near-perfect alignment indicates that OSA effectively places almost all noisy samples behind the clean ones.

Table 6: ACC and recall of noise detection.

| Estimator Type | Noise Ratio | Acc | Recall |
|---|---|---|---|
| CLIP (w/o DA) | 0.2 | 93.88 | 97.49 |
| CLIP (w DA) | 0.2 | 97.68 | 97.18 |
| CLIP (w/o DA) | 0.5 | 93.91 | 99.35 |
| CLIP (w DA) | 0.5 | 98.14 | 99.24 |

Table 7: Overhead Comparison.

| Model | Time | Extra Time |
|---|---|---|
| CLIP | 97 min | 0 min |
| NPC | 323 min | 226 min |
| OSA | 118 min | 21 min |

## 4.6 COMPUTATIONAL COST ANALYSIS

**Cost in Pre-training.**    To evaluate the practicality of OSA in a real-world pre-training scenario, we estimate the additional computational cost for processing 1 billion data points. Using an NVIDIA RTX 3090 with an inference batch size of 4096, utilizing approximately 24 GB of GPU memory, processing the MS-COCO dataset consisting of 566,435 pairs takes approximately 153 seconds. At this inference rate, processing 1 billion data points would require approximately 75 hours on a single RTX 3090. This cost is negligible within the context of large-scale pre-training, especially when leveraging multiple GPUs for parallel inference.

**Time Cost Comparison.**    To further examine the computational efficiency of our method compared to other anti-noise techniques, we evaluate training time against two representative approaches: CLIP and NPC. CLIP, which serves as the baseline, is trained directly without any additional technique. NPC, the current state-of-the-art, also uses CLIP as its backbone but applies an anti-noise technique by estimating the negative impact of each sample, necessitating double backward passes. The training time comparison, presented in Table. 7, shows that our method introduces only a minimal increase in training time compared to direct training, requiring just one-tenth of the additional time needed by NPC. This highlights the efficiency of OSA, making it well-suited for large-scale robust training tasks.

## 5 CONCLUSION

**Broader Impacts.**    In this work, we investigated the possibility of anti-noise in practical large-scale training. We introduced a novel model-agnostic anti-noise paradigm with advantages such as task transferability, model adaptability, and low computational overhead. By leveraging the properties of high-dimensional spaces, we found a robust and effective boundary for distinguishing between noisy and clean samples. Through rigorous theoretical analysis and comprehensive experimentation, we validated the efficacy and robustness of OSA for general noise mitigation. Although our primary objective is to adapt to practical large-scale training, OSA also achieves SOTA performance in standard anti-noise settings. To the best of our knowledge, this is the first work to explore noise mitigation in practical large-scale training scenarios, as well as the first to propose a general anti-noise approach.

**Limitations and Future Works.**    Limited by the significant computational cost of pre-training, it is difficult for us to evaluate in a real pre-training process. Instead, we simulate large-scale pre-training processes to the greatest extent possible, such as evaluating on the real-world noisy dataset CC120K, which shares similar domains with mainstream pre-training datasets like CC4M and CC12M. Exploring the broad domain adaptability of OSA in real pre-training scenarios will be a valuable direction for future work.

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

APPENDIX

# A    DETAILS OF IMPLEMENTATION

**Dataset Details.**    MSCOCO is widely used for noisy cross-modal matching, with each image accompanied by five descriptive captions. Following the setting of Huang et al. (2021), we utilize 113,287 images for training, 5,000 for validation, and 5,000 for testing. The Flickr30K dataset encompasses 31,783 image-text instances, each image paired with five textual annotations. Adhering to the NCR (Huang et al., 2021), we use 29,783 images for training and 1,000 images each for validation and testing. Regarding noise splits, following the NCR categorization, we conduct experiments at noise ratios of 0%, 20%, 40%, and 60%. CC120K is a real-world multimodal noisy dataset collected by Zhang et al. (2024) from the Internet, with about 3%-20% noise ratio. There are 118,851 image-text pairs for training, 1,000 for validation, and 1,000 for testing.

The Aircraft, Bird, and Car we used in the image classification task are three non-overlapping subsets of the WebFG-496 (Sun et al., 2021) dataset. WebFG-496 consists of 53,339 images, totaling 496 subcategories. This dataset is annotated using a webly supervised approach, which leverages resources from web search engines (*e.g.*, Google Image Search Engine, Bing Image Search Engine) to expand the annotated image dataset.

For the image retrieval task, we conduct experiments on the CARS98N dataset under PRISM's setting (Liu et al., 2021). We utilize 9,558 car-related images sourced from text-based searches on Pinterest as the training set, and employ the remaining 98 categories from CARS, unsearched on Pinterest, as a clean test set. The dataset's noise is inherently real-world, with its creators estimating a noise ratio of approximately 50%.

**Implementation Details.**    To demonstrate the effectiveness of the OSA, we incorporate an estimator, built around the core of CLIP, and re-weighting operations based on the Estimator's outcomes into numerous downstream tasks. In the principal task of cross-modal image-text retrieval, we employ CLIP with ViT-B/32 as the baseline and target model by default. All experiments are conducted on a single RTX 3090 GPU using the AdamW optimizer. During both training phases, the model is trained for five epochs with a batch size of 256 and 500 warmup steps.

For the image classification task on the WebFG dataset, we align with the field's prevalent models for a fair comparison by employing the ResNet-50 model enhanced by CLIP for feature extraction and the CLIP image encoder as our estimator. Training and testing are executed on single RTX 3090 GPU, with an input image resolution of $448 \times 448$. The batch size and initial learning rate are specified as 64 and 1e-5, respectively. In the first phase, the estimator is trained with data modeled by a Gaussian Mixture Model (GMM), which considers the classification and matching losses of all training samples, with the GMM probability threshold of 0.95. The classification task leverages the CLIP protocol, where a fixed prompt ("This is a picture of") is prepended to category texts.

For the image retrieval task, we use CLIP ViT-B/32 as the baseline, with a batch size set to 128, an initial learning rate of 5e-6, and the number of epochs set to 10. Following the setup of the PRISM (Liu et al., 2021), we set the parameter for sampling positive examples by the random sampler of the dataloader to 4, and adjust the number of positive examples sampled per epoch to one-fourth of the original parameter according to the increase in batch size. In this task, we also adopt a two-stage training approach. The strictly clean in-domain training data for the first stage is obtained using a GMM model with a probability setting of 0.8.

# B    RELATED WORK

## B.1    NOISE MITIGATION IN CROSS-MODAL MATCHING

The cross-modal matching task (Lee et al., 2018; Song & Soleymani, 2019; Li et al., 2019; 2022; Diao et al., 2021) serves as a fundamental component in multimodal learning. However, the inherent difference in information density between these modalities leads to high annotation costs and inconsistent annotation quality, rendering cross-modal tasks particularly vulnerable to label noise. Some approaches explicitly identify and correct noisy samples through cross-prediction between

concurrently trained dual models (Huang et al., 2021; Yang et al., 2023a; Liang et al., 2022b), while others (Zhang et al., 2024; Qin et al., 2022) implicitly estimate the probability of sample noise, reducing its training impact by adjusting the loss function. NCR (Huang et al., 2021) employs the memorization capacity of its counterpart model for simple clean samples to rectify the output results. BiCro (Yang et al., 2023a) utilizes the consistency of similarity score distributions from a Siamese model ensemble on noisy data, alongside anchors modeled on the loss distribution via a Beta-Mixture-Model (BMM), to filter out noisy samples. NPC (Zhang et al., 2024), deviating from the dual-model training schemes, introduces a two-stage single-model training approach that reduces training overhead by replacing two backward passes with one forward and one backward pass. Specifically, the first stage estimates the impact of potentially noisy samples on model performance by constructing a high-quality clean sample bank; the second stage then utilizes these estimates to reweight the loss function. However, current methods for distinguishing clean from noisy samples rely on numerous hyperparameters that are closely linked to dataset size and model capacity. This dependency not only limits their adaptability to various downstream tasks but also makes them challenging to deploy in real-world applications.

## B.2 Noise Mitigation in Image Classification

Image classification is vulnerable to training data noise, due to varied noise types and strong model memorization. Noise in datasets manifests in two primary forms: synthetic alterations and those arising from real-world scenarios. The former typically involves shuffling the labels of a subset of the data or retaining the labels while introducing corresponding category images from external datasets. The latter entails substituting images for a random selection of data points with those sourced from image search engines. Existing approaches are categorized based on their operational focus: loss correction (Yi & Wu, 2019; Zhang & Sabuncu, 2018a; Menon et al., 2015; Natarajan et al., 2013; Patrini et al., 2017; Xia et al., 2019; Ghosh et al., 2017; Wang et al., 2019a;b; Xu et al., 2019; Zhang & Sabuncu, 2018b) and sample selection (Sun et al., 2022; Albert et al., 2023; Yao et al., 2021; Li et al., 2020; Albert et al., 2022). Loss correction methods typically incorporate a regularization term into the loss function, implicitly reweighting clean and noisy samples within the loss. Sample selection strategies, in contrast, explicitly differentiate between clean and noisy samples, applying distinct processing to each category during loss computation. Representative for the loss correction category, (Zhang & Sabuncu, 2018a) aims to generalize ordinary Cross-Entropy loss and MAE loss by setting the loss threshold to iid and ood noisy samples. DivideMix (Li et al., 2020) concurrently trains two networks, each utilizing the data partitioning from the other network to distinguish between clean and noisy samples based on loss values, thereby mitigating the influence of confirmation bias inherent within each network. PNP (Sun et al., 2022) framework employs a unified predictive network to estimate the in-distribution (iid), out-of-distribution (ood), and clean probabilities for a given sample. Co-training trained on a sample that has a lower loss, and with the different predictions by its siamese network.

## B.3 Noise Mitigation in Image Retrieval.

Although image retrieval tasks focus on pairwise relationships, the noise predominantly originates from image categorization errors. Analogous to image classification tasks, this can be bifurcated into in-domain (Wang & Tan, 2018) and open-set noise (Liu et al., 2021). In terms of task configuration, noise retrieval typically operates at the category level, treating images within the same category as positive instances. PRISM (Liu et al., 2021) tries to find noisy image samples by finding the outliers score in the whole similarity matrix from the same category. The generalization ability of the image feature is ensured by a broader query bank restored multi-view of it. TITAN (Yang et al., 2023b) utilizes prototypes to be representative of the anchor of the clean and noisy samples and then generates synthetic samples by a combination of prototypes for substitution of noisy samples. T-SINT (Ibrahimi et al., 2022b) utilizes more negative samples by the interaction between noisy samples and negative samples that belong to another category.

## C  Proofs

### C.1  Proof of High-dimensional Orthogonality

Suppose $u, v \in \mathbb{R}^d$ are any two random vectors. The cosine similarity $\cos(u, v) \sim \mathcal{N}(0, d^{-1})$. The probability that $\cos(u, v)$ is within a specific range $[-a, a]$ is denoted as:

$$P(-a \leq \cos(u, v) \leq a) = \Phi\left(\frac{a}{\varsigma}\right) - \Phi\left(\frac{-a}{\varsigma}\right), \tag{7}$$

where $\Phi$ represents the CDF of the standard normal distribution, and $\varsigma = \frac{1}{\sqrt{d}}$ is the standard deviation of the cosine similarity. When $d = 1024$ and $a = 0.1$, there are

$$\varsigma = \frac{1}{\sqrt{1024}} = \frac{1}{32}, \tag{8}$$

and

$$P(-0.1 \leq \cos(u, v) \leq 0.1) = \Phi\left(\frac{0.1}{1/32}\right) - \Phi\left(\frac{-0.1}{1/32}\right) \approx 0.9986. \tag{9}$$

### C.2  Proof of Theorem 1

In the Section. 2.2, we propose that Theorem 1 about the relative relationship of pairs in the original entire space, will not change after transmitting to the narrow cone space of the trained model, and there is always a boundary $r$ concentrated on most random vectors.

To prove this Theorem, we first introduce a useful lemma of monotonicity of cosine similarity proposed by Liang et al. (2022a), indicating that the cosine similarity between two vectors increases with a high probability after one feedforward computation consisting of a linear transformation and ReLU computation.

**Lemma 1.** *Suppose $u, v \in \mathbb{R}^{d_{in}}$ are any two fixed vectors such that $\|u\| = r\|v\|$ for some $r > 0$, $\mathbf{W} \in \mathbb{R}^{d_{out} \times d_{in}}$ is a random weight matrix where each element $\mathbf{W}_{k,l} \sim \mathcal{N}(0, d_{out}^{-1})$ for $k \in [d_{out}]$, $l \in [d_{in}]$, and $\mathbf{b} \in \mathbb{R}^{d_{out}}$ is a random bias vector such that $\mathbb{b}_k \sim \mathcal{N}(0, d_{out}^{-1})$ for $k \in [d_{out}]$. If $\cos(u, v) < (\frac{1}{2}(r + \frac{1}{r}))^{-1}$, then the following holds with probability at least $1 - O(1/d_{out})$.*

$$\cos(\sigma(\mathbf{W}u + \mathbf{b}), \sigma(\mathbf{W}v + \mathbf{b})) > \cos(u, v). \tag{10}$$

*Proof of Theorem. 1.* Let $\mathbb{R}^{d_{in}}$ be the original space before being transmitted in a neural network. Suppose $u, v \in \mathbb{R}^{d_{in}}$ are any two random vectors with $\cos(u, v) \approx 0$. $u_c, v_c \in \mathbb{R}^{d_{in}}$ is a pair of clean vectors with $\cos(u_c, v_c) > 0$, while $u_n, v_n \in \mathbb{R}^{d_{in}}$ is a noisy pair with $\cos(u_n, v_n) < 0$. Given a Neural Network $F(x) = f_t(f_{t-1}(\ldots f_2(f_1(x)))) \in \mathbb{R}^{d_{out}}$ with $t$ layers. $f_i(x) = \sigma_i(\mathbf{W}_i x + \mathbf{b}_i)$ denotes $i^{th}$ layer, where $\sigma(\cdot)$ indicates activation function. $\mathbf{W}_i \in \mathbb{R}^{d_{out}^i \times d_{in}^i}$ is a random weight matrix where each element $\mathbf{W}_i^{k,l} \sim \mathcal{N}(0, 1/d_{out}^i)$ for $k \in \left[d_{out}^i\right]$, $l \in \left[d_{in}^i\right]$, and $\mathbf{b}_i \in \mathbb{R}^{d_{out}^i}$ is a random bias vector such that $\mathbf{b}_i^k \sim \mathcal{N}(0, 1/d_{out}^i)$ for $k \in \left[d_{out}^i\right]$. We would like to prove that there are always be a boundary $\beta$, satisfying:

$$\cos(F(u_n), F(v_n)) < \cos(F(u), F(v)) \approx \beta < \cos(F(u_c), F(v_c)), \tag{11}$$

which is equivalent to proving.

$$\cos(f_i(u_n), f_i(v_n)) < \cos(f_i(u), f_i(v)) \approx \beta_i < \cos(f_i(u_c), f_i(v_c)), \tag{12}$$

where $\beta_i$ is the boundary of $i^{th}$ layer.

We first consider the cosine similarity between $u$ and $v$ as:

$$\cos(u, v) = \frac{u \cdot v}{\|u\|\|v\|}. \tag{13}$$

After a linear transformation of $i^{th}$ layer, the cosine similarity of $\cos(\mathbf{W}_i u + \mathbf{b}_i, \mathbf{W}_i v + \mathbf{b}_i)$ denotes:

$$\cos(\mathbf{W}_i u + \mathbf{b}_i, \mathbf{W}_i v + \mathbf{b}_i) = \frac{(\mathbf{W}_i u + \mathbf{b}_i) \cdot (\mathbf{W}_i v + \mathbf{b}_i)}{\|\mathbf{W}_i u + \mathbf{b}_i\|\|\mathbf{W}_i v + \mathbf{b}_i\|}. \tag{14}$$

Since $\mathbf{b}_i$ has a mean of zero and is independent from $\mathbf{W}_i u$ and $\mathbf{W}_i v$, the expectation of $\mathbf{b}_i$ and $(\mathbf{W}_i u + \mathbf{b}_i) \cdot \mathbf{W}_i v + \mathbf{b}_i)$ can be signified as:

$$\mathbb{E}\left[\mathbf{b}_i\right] = 0, \tag{15}$$

$$\mathbb{E}\left[(\mathbf{W}_i u + \mathbf{b}_i) \cdot (\mathbf{W}_i v + \mathbf{b}_i)\right] = \mathbb{E}\left[(\mathbf{W}_i u \cdot \mathbf{W}_i v)\right] = \sum_{i=1}^{n}\sum_{i=1}^{n}\frac{1}{d_{out}^i}u_k v_k = \frac{1}{d_{out}^i}(u \cdot v). \tag{16}$$

Additionally, we have

$$\|\mathbf{W}_i u + \mathbf{b}_i\|^2 = \mathbf{W}_i u \cdot \mathbf{W}_i u + 2\mathbf{W}_i u \cdot \mathbf{b}_i + \mathbf{b}_i \cdot \mathbf{b}_i. \tag{17}$$

Due to $\mathbb{b}^k \sim \mathcal{N}(0, 1/d_{out}^i)$, as $d_{o}^i ut$ increases, the term of $2\mathbf{W}_i u \cdot \mathbf{b}_i$ and $\mathbf{b}_i \cdot \mathbf{b}_i$ become negligible, which implies

$$\|\mathbf{W}_i u + \mathbf{b}_i\|^2 \approx \mathbf{W}_i u \cdot \mathbf{W}_i u = \sum_{i=1}^{n}(\mathbf{W}_i u)^2. \tag{18}$$

Therefore, the expectation of $\|\mathbf{W}_i u + \mathbf{b}_i\|^2$ is approximate to

$$\mathbb{E}\left[\|\mathbf{W}_i u\|^2\right] = \sum_{k=1}^{n} u_k^2 \frac{1}{d_{out}^i} = \frac{\|u\|}{d_{out}^i}, \tag{19}$$

and

$$\begin{aligned}\cos(\mathbf{W}_i u + \mathbf{b}_i, \mathbf{W}_i v + \mathbf{b}_i) &\approx \frac{\mathbb{E}[\mathbf{W}_i u \cdot \mathbf{W}_i v]}{\sqrt{\mathbb{E}[\|\mathbf{W}_i u + \mathbf{b}_i\|^2]\mathbb{E}[\|\mathbf{W}_i v + \mathbf{b}_i\|^2]}} \\ &= \frac{\frac{1}{d_{out}^i}(u \cdot v)}{\sqrt{\frac{1}{d_{out}^i}\|u\|^2 \cdot \frac{1}{d_{out}^i}\|v\|^2}} \\ &= \cos(u, v).\end{aligned} \tag{20}$$

Based on Eq. 20, with $\cos(u_n, v_n) < \cos(u, v) \approx 0 < \cos(u_c, v_c)$, there are

$$\cos(\mathbf{W}_i u_n + \mathbf{b}_i, \mathbf{W}_i v_n \mathbf{b}_i) < \cos(\mathbf{W}_i u + \mathbf{b}_i, \mathbf{W}_i v + \mathbf{b}_i) < \cos(\mathbf{W}_i u_c + \mathbf{b}_i, \mathbf{W}_i v_c + \mathbf{b}_i). \tag{21}$$

Since the activation function $\sigma$ is a monotonically increasing function, it follows

$$\cos(f_i(u_n), f_i(v_n)) < \cos(f_i(u), f_i(v)) < \cos(f_i(u_c), f_i(v_c)). \tag{22}$$

Due to Lemma. 1, $\cos(f_i(u), f_i(v))$ will be increase with the transmitting layers, and $\cos(f_i(u), f_i(v))$ will always be a $\beta_i > 0$, to satisfy:

$$\cos(f_i(u_n), f_i(v_n)) < \cos(f_i(u), f_i(v)) \approx \beta_i < \cos(f_i(u_c), f_i(v_c)). \tag{23}$$

After transmitting each layer, Eq. 23 are always satisfied. When transmitting a neural network with $t$ layers, we have

$$\cos(F(u_n), F(v_n)) < \cos(F(u), F(v)) \approx \beta < \cos(F(u_c), F(v_c)). \tag{24}$$

$\square$

## C.3 PROOF OF ORTHOGONALITY VALIDITY IN CONE SPACE

Although we have demonstrated in Appendix. C.1 that in the original high-dimensional space, the cosine similarity between two randomly selected vectors—each dimension following a Gaussian distribution—typically converges near the orthogonal boundary, this property may not necessarily extend to the subspace of the shared embedding space maintained by the trained models. Specifically, for real image-text pairs, the subspace may deviate from the orthogonal characteristics observed in the original space. Thus, it is essential to investigate whether the orthogonality property holds within the cone space for the image-text subdomain post-training.

To explore this, we first analyze the distribution of several dimensions of image and text features from the CC120K dataset, as illustrated in Figure. 3. The results reveal that all vector dimensions,

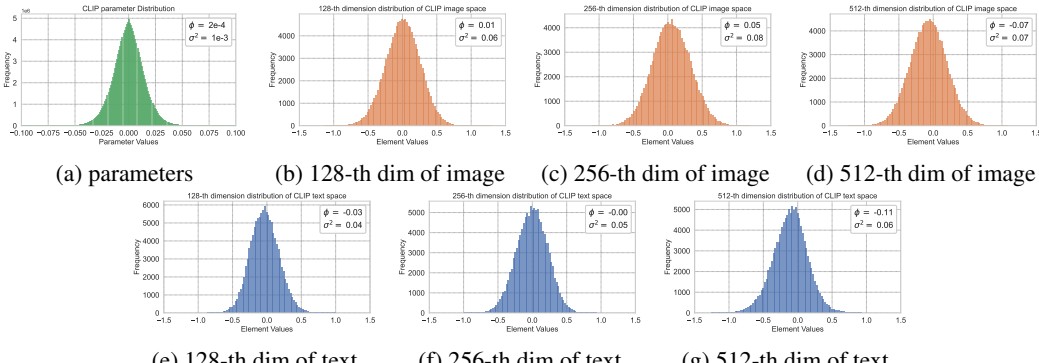

(a) parameters     (b) 128-th dim of image     (c) 256-th dim of image     (d) 512-th dim of image

(e) 128-th dim of text     (f) 256-th dim of text     (g) 512-th dim of text

Figure 3: The illustrations of several distributions on CC120K. (a) The parameter distribution. (b-d) The distribution of image features for the 128th, 256th, and 512th dimensions. (e-g) The distribution of text features for the 128th, 256th, and 512th dimensions.

including trained parameters, exhibit a Gaussian distribution with near-zero means. If the dimensions of the trained embedding space follow Gaussian distributions, the process of selecting random vectors within this space would be analogous to that of the original space, thereby preserving the orthogonality property. Here, we present the following theorem: The output features of large-scale models tend to Gaussian distribution. The detailed theorem and proof are provided below.

**Theorem 2** (Output features tends to Gaussian). *Given a Neural Network $F(x) = \{f_t(f_{t-1}(\ldots f_2(f_1(x))))\} \in \mathbb{R}^{d_{out}}$ with $t$ layers. $f_l(x) = \phi_l(\mathbf{W}_l x + \mathbf{b}_l)$ denotes the $l^{th}$ layer, where $\phi(\cdot)$ indicates the activation function, and the final layer $f_t(x) = \mathbf{W}_t x + \mathbf{b}_t$ is a fully-connected layer without an activation function for common space projection. Let $x^k \in \mathbb{R}^{d_{in}^k}$ be the sample feature that will be transmitted into the $k^{th}$ layer, where $x^1$ denotes the original feature with an unknown distribution $x^1 \sim (\mu_x, \sigma_x^2)$. $\mathbf{W}_k \in \mathbb{R}^{d_{out}^k \times d_{in}^k}$ is a random weight matrix where each element $w_{ij}^k \sim \mathcal{N}(0, \sigma_w^2)$ for $i \in [d_{out}^k]$, $j \in [d_{in}^k]$, and $\mathbf{b}_k \in \mathbb{R}^{d_{out}^k}$ is a bias vector such that $b_i^k \sim \mathcal{N}(0, \sigma_w^2)$ for $i \in [d_{out}^k]$. In such a neural network, linear layers lead features $x$ gradually to a Gaussian distribution from any initial distribution, and as $|d_{in}|$ is sufficiently large, $F(x) \sim \mathcal{N}(0, \sigma^2)$.*

*Proof of Theorem.* 2. For the $k^{th}$ layer ($k \in [t]$), we first calculate the expectation and variance of the linear combination $\sum_{j=1}^{d_{in}^k} w_{ij}^k x_j^k$. For the expectation, since $w_{ij}^k$ and $x_j^k$ are independent and $\mathbf{w}_{ij}^k \sim \mathcal{N}(0, \frac{1}{d_{out}^k})$, we have:

$$\mathbb{E}\left[\sum_{j=1}^{d_{in}^k} w_{ij}^k x_j^k\right] = \sum_{j=1}^{d_{in}^k} \mathbb{E}[w_{ij}^k]\mathbb{E}[x_j^k] = \sum_{j=1}^{d_{in}^k}(0 \times \mathbb{E}[x_j^k]) = 0. \tag{25}$$

For variance, since $w_{ij}^k$ and $x_j^k$ are independent, we have:

$$\begin{aligned}
\mathrm{Var}\left(\sum_{j=1}^{d_{in}^k} w_{ij}^k x_j^k\right) &= \sum_{j=1}^{d_{in}^k} \mathrm{Var}(w_{ij}^k x_j^k) = \sum_{j=1}^{d_{in}^k} \mathbb{E}\left[(w_{ij}^k)^2 (x_j^k)^2\right] \\
&= \sum_{j=1}^{d_{in}^k} \mathbb{E}\left[(w_{ij}^k)^2\right] \mathbb{E}\left[(x_j^k)^2\right] \\
&= \sum_{j=1}^{d_{in}^k} \sigma_{w^k}^2 \left(\mathrm{Var}(x_j^k) + (\mathbb{E}[x_j^k])^2\right) \\
&= \sum_{j=1}^{d_{in}^k} \sigma_{w^k}^2 \left(\sigma_{x^k}^2 + \mu_{x^k}^2\right) = d_{in}^k \sigma_{w^k}^2 \left(\sigma_{x^k}^2 + \mu_{x^k}^2\right).
\end{aligned} \tag{26}$$

Since $w_{ij}^k$ are independently distributed Gaussian random variables, and $x_j^k$ has a known mean and variance, the sum of $w_{ij}^k x_j^k$ can apply to a generalized Central Limit Theorem. We have

$$\frac{\sum_{j=1}^{d_{in}^k} w_{ij}^k x_j^k - \mathbb{E}\left[\sum_{j=1}^{d_{in}^k} w_{ij}^k x_j^k\right]}{\sqrt{\text{Var}\left(\sum_{j=1}^{d_{in}^k} w_{ij}^k x_j^k\right)}} \xrightarrow{d} \mathcal{N}(0,1), \tag{27}$$

which is equivalent to

$$\frac{\sum_{j=1}^{d_{in}^k} w_{ij}^k x_j^k - 0}{\sqrt{d_{in}^k \sigma_{w^k}^2 (\sigma_{x^k}^2 + \mu_{x^k}^2)}} \xrightarrow{d} \mathcal{N}(0,1). \tag{28}$$

Therefore,

$$\sum_{j=1}^{d_{in}^k} w_{ij}^k x_j^k \xrightarrow{d} \mathcal{N}(0, d_{in}^k \sigma_{w^k}^2 (\sigma_{x^k}^2 + \mu_{x^k}^2)). \tag{29}$$

Due to $b^k \sim \mathcal{N}(0, \sigma_b^2)$, we finally get

$$\sum_{j=1}^{d_{in}^k} w_{ij}^k x_j^k + b_i^k \xrightarrow{d} \mathcal{N}\left(0, d_{in}^k \sigma_{w^k}^2 (\sigma_{x^k}^2 + \mu_{x^k}^2) + \sigma_b^2\right). \tag{30}$$

Although activation functions truncate the Gaussian distribution after each linear layer, the samples still gradually approach a Gaussian distribution from the initial unknown distribution as they pass through the layers. Furthermore, because there is a fully connected layer ( layer) without an activation function before mapping to the final common space, the final feature distribution will approximate a Gaussian distribution, as follows:

$$F(x) \sim \mathcal{N}(0, d_{in}^t \sigma_{w^t}^2 (\sigma_{x^t}^2 + \mu_{x^t}^2) + \sigma_b^2). \tag{31}$$

$\square$

## D SDM VISUALIZATION

We visualize some representative samples from our synthetic domain originating from COCO by using SDM. The results are shown in Figure. 4. We generate two styles of image based on the MSCOCO caption, and then use pre-trained multimodal models to calculate cosine similarity with the SDM-generated image and original caption.

**Sketch Style**

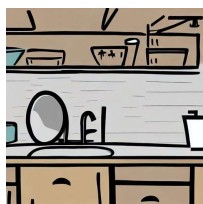 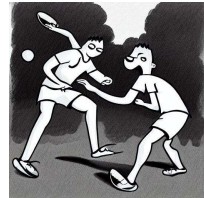 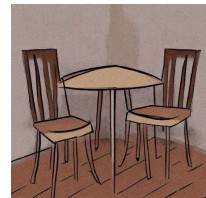

A kitchen counter with dirty dishes near sink.

A couple of young men playing a game of frisbee.

A wooden kitchen table with three wooden chairs.

**Cartoon Style**

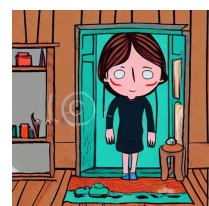 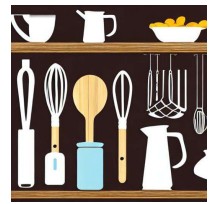 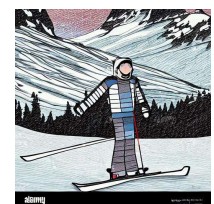

A woman in a young girl inside of the cabin.

Miscellaneous kitchen utensils on a wooden shelf.

Skier standing for picture on a gentle slope.

Figure 4: Examples of generated SDM dataset. The first row is in sketch style, while the second row is in cartoon style.

# E  ADDITIONAL EXPERIMENTAL RESULTS

Table 8: Comparison on noisy Flickr30K.

| Method | Noise ratio | i2t R@1 | R@5 | R@10 | t2i R@1 | R@5 | R@10 | Noise ratio | i2t R@1 | R@5 | R@10 | t2i R@1 | R@5 | R@10 |
|--------|-------------|---------|-----|------|---------|-----|------|-------------|---------|-----|------|---------|-----|------|
| NCR | | 77.3 | 94.0 | 97.5 | 59.6 | 84.4 | 89.9 | | 73.5 | 93.2 | 96.6 | 56.9 | 82.4 | 88.5 |
| DECL | | 79.8 | 94.9 | 97.4 | 59.5 | 83.9 | 89.5 | | 77.5 | 93.8 | 97.0 | 56.1 | 81.8 | 88.5 |
| BiCro | 0% | 81.7 | 95.3 | 98.4 | 61.6 | 85.6 | 90.8 | 20% | 78.1 | 94.4 | 97.5 | 60.4 | 84.4 | 89.9 |
| NPC | | 87.9 | **98.1** | **99.4** | 75.0 | **93.7** | **97.2** | | 87.3 | 97.5 | 98.8 | 72.9 | 92.1 | 95.8 |
| CLIP | | 86.2 | 97.6 | 99.2 | 72.9 | 92.3 | 96.0 | | 82.3 | 95.5 | 98.3 | 66.0 | 88.5 | 93.5 |
| +OSA | | **88.6** | 97.7 | 99.3 | **75.6** | 93.6 | 96.8 | | **88.9** | **97.7** | **99.1** | **75.6** | **93.3** | **96.9** |
| NCR | | 68.1 | 89.6 | 94.8 | 51.4 | 78.4 | 84.8 | | 13.9 | 37.7 | 50.5 | 11.0 | 30.1 | 41.4 |
| DECL | | 72.7 | 92.3 | 95.4 | 53.4 | 79.4 | 86.4 | | 65.2 | 88.4 | 94.0 | 46.8 | 74.0 | 82.2 |
| BiCro | 40% | 74.6 | 92.7 | 96.2 | 55.5 | 81.1 | 87.4 | 60% | 67.6 | 90.8 | 94.4 | 51.2 | 77.6 | 84.7 |
| NPC | | 85.6 | 97.5 | 98.4 | 71.3 | 91.3 | 95.3 | | 83.0 | 95.9 | 98.6 | 68.1 | 89.6 | 94.2 |
| CLIP | | 76.2 | 93.3 | 96.5 | 59.4 | 85.0 | 90.9 | | 66.3 | 87.3 | 93.0 | 52.1 | 78.8 | 87.4 |
| +OSA | | **87.3** | **97.6** | **99.3** | **74.2** | **93.1** | **96.7** | | **87.2** | **98.1** | **99.6** | **74.4** | **92.9** | **96.4** |

Table 9: The results of other methods employing OSA on MSCOCO 1K.

| Noise Ratio | Method | i2t | | | t2i | | |
|---|---|---|---|---|---|---|---|
| | | R@1 | R@5 | R@10 | R@1 | R@5 | R@10 |
| 0% | NPC | 82.2 | **96.5** | **98.7** | 68.3 | **92.0** | **98.7** |
| | **+OSA** | **82.4** | 96.4 | 98.6 | **68.5** | 91.8 | **98.7** |
| 20% | NPC | 79.9 | 95.9 | 98.4 | 66.3 | 90.5 | 98.4 |
| | **+OSA** | **81.2** | **96.0** | **98.6** | **66.9** | **91.2** | **98.6** |
| 50% | NPC | 78.2 | 94.4 | 97.7 | 63.1 | 89.0 | 97.7 |
| | **+OSA** | **79.3** | **95.6** | **98.2** | **66.8** | **90.8** | **98.2** |

Table 10: Ablation study of estimator type on noisy MS-COCO.

| Noise ratio | Estimator | MS-COCO 1K | | | | | | MS-COCO 5K | | | | | |
|---|---|---|---|---|---|---|---|---|---|---|---|---|---|
| | | i2t | | | t2i | | | i2t | | | t2i | | |
| | | R@1 | R@5 | R@10 | R@1 | R@5 | R@10 | R@1 | R@5 | R@10 | R@1 | R@5 | R@10 |
| 0% | None | 80.1 | 95.7 | 98.2 | 67.1 | 91.4 | 96.6 | 62.9 | 84.9 | 91.6 | 46.5 | 73.8 | 82.9 |
| | CLIP (w/o DA) | **82.6** | **96.7** | 98.7 | 68.5 | **92.1** | **96.7** | **66.2** | **87.0** | **93.3** | 48.6 | 75.7 | **84.8** |
| | ALIGN (w/o DA) | 81.9 | 96.7 | 98.7 | 68.9 | 92.2 | 96.9 | 64.8 | 86.6 | 92.7 | 49.0 | 75.9 | 84.7 |
| | **CLIP (w DA)** | 82.2 | 96.5 | 98.7 | **68.8** | 92.1 | 96.7 | 65.6 | 86.8 | 92.9 | **49.1** | **76.2** | **84.8** |
| 20% | None | 76.0 | 94.3 | 97.5 | 63.4 | 89.0 | 94.8 | 55.3 | 79.1 | 86.9 | 41.0 | 68.8 | 79.3 |
| | CLIP (w/o DA) | **81.8** | 96.1 | **98.7** | 68.2 | 91.9 | 96.5 | 64.8 | **86.6** | 92.3 | 48.3 | 75.4 | 84.1 |
| | ALIGN (w/o DA) | 81.2 | 96.0 | 98.6 | 67.7 | 91.5 | 96.4 | 64.8 | 86.2 | 92.3 | 47.8 | 74.9 | 83.9 |
| | **CLIP (w DA)** | 81.6 | **96.2** | 98.5 | **68.9** | **92.0** | **96.6** | **65.8** | 86.4 | **92.5** | 48.7 | 76.1 | 84.5 |
| 50% | None | 73.9 | 93.0 | 97.2 | 60.1 | 87.3 | 94.0 | 54.1 | 78.5 | 86.6 | 39.7 | 67.2 | 77.5 |
| | CLIP (w/o DA) | 79.6 | 95.6 | 98.4 | 65.9 | 90.8 | 95.9 | 62.4 | 84.8 | 90.8 | 45.7 | 73.1 | 82.5 |
| | ALIGN (w/o DA) | 80.4 | 95.6 | 98.3 | 66.0 | 90.5 | 95.8 | 62.0 | 84.9 | 91.8 | 45.7 | 73.2 | 82.5 |
| | **CLIP (w DA)** | 80.4 | **96.2** | **98.6** | 67.8 | **91.6** | **96.4** | **64.0** | **85.5** | **91.9** | 47.9 | 74.6 | 83.8 |

Table 11: Mean Noise Rank Comparison between OSA and NPC.

| Noise Ratio | Method | Mean Noise Rank↑ | Optimal Rank | Noise Number | Sample Number |
|---|---|---|---|---|---|
| 20% | NPC | 1641.3 | 1815.5 | 370 | 2,000 |
| | OSA | **1809.1** | 1815.5 | 370 | 2,000 |
| 50% | NPC | 1456.2 | 1524.0 | 953 | 2,000 |
| | OSA | **1520.7** | 1524.0 | 953 | 2,000 |

