# OpenReview forum: "One-step Noisy Label Mitigation"
_ICLR.cc/2025/Conference — ICLR 2025 Conference Withdrawn Submission_

### Official Review · Reviewer_7z7z · 2024-10-30

**Soundness:** 2
**Presentation:** 3
**Contribution:** 2
**Rating:** 5
**Confidence:** 3

**Summary:**

This paper focuses on how to mitigate noisy labels, in particular noisy cross-modal matching. Specifically, the authors first use a pre-trained model, such as CLIP, to determine whether a data is noisy, and then give less weight to the noisy label during the training process. And they pointed out that the orthogonal boundary separates the clean and noisy sides. The authors conducted experiments on different tasks and datasets.

**Strengths:**

1. The writing and presentation of the paper is clear.
2. The boundary principle analysis of the paper is  instructive.
3. The experiments in this paper are detailed and show validity.

**Weaknesses:**

1. I think it would be better if the authors emphasised in the title or elsewhere that the proposed work focuses primarily on noisy cross-modal matching. Otherwise it could be confusing. For example, the authors claim that other methods cause additional computational overhead. However, papers cited by the authors in related work, such as [1], do not incur additional overhead; rather, the proposed work causes additional overhead.
2. The paper doesn't seem to describe how big the CLIP is as an Estimator. If the author uses a trained maximum CLIP as an Estimator, then of course there will be a performance boost because it is a strong model. That doesn't seem fair to the baselines.
3. An approach that relies on trained large models does not seem very interesting. And regarding Eq. 5, the authors do not provide a theoretical analysis.

[1] Generalized Cross Entropy Loss for Training Deep Neural Networks with Noisy Labels, NeurIPS 2018

**Questions:**

1. The paper does not seem to describe whether the backbone in the experiment was randomly initialized or trained. As I understand it, the estimator is a trained CLIP and the backbone for the baselines is also a trained CLIP. Is this correct?

---

### Official Review · Reviewer_Hf2c · 2024-11-02

**Soundness:** 2
**Presentation:** 1
**Contribution:** 2
**Rating:** 5
**Confidence:** 5

**Summary:**

This paper focuses on the very practical problem of noisy labels in the dataset. The authors propose a sample weighting mechanism based on pre-trained models, especially visual language models such as CLIP.

**Strengths:**

1. The problem studied in this paper is very important, especially in the current era when large models are so popular.
2. Though it has been discussed and proposed before, it is reasonable to use additional models to help with sample selection and reweighting.

**Weaknesses:**

1. The paper is difficult to read, primarily because it lacks a clear problem definition section. For instance, while I understand the intuitive idea, how is "cleanness" mathematically defined? Should we assume that \( x \) and \( y \) come from a shifted joint distribution? Then, how is the noisy distribution structured, and what type of noise is being used? Additionally, what does the noise ratio in the experiments represent? For instance, in the COCO dataset, did you randomly replace a proportion of captions? This lack of clarity also makes it hard for me to understand the significance of Theorem 1 and the related analysis.

2. The paper lacks a discussion of important related literature. I would list a few representative methods in learning with noisy labels community:
   - *[1]* DivideMix: Learning with Noisy Labels as Semi-supervised Learning.

   And sample selection methods based on feature space, which are more relevant to this work:
   - *[2]* Multi-Objective Interpolation Training for Robustness to Label Noise
   - *[3]* FINE Samples for Learning with Noisy Labels

   There are also papers that use the CLIP model:
   - *[4]* CLIPCleaner: Cleaning Noisy Labels with CLIP
   - *[5]* Vision-Language Models are Strong Noisy Label Detectors
   - *[6]* Combating Label Noise With A General Surrogate Model For Sample Selection

   (*Some of these references may be considered concurrent work; The authors are suggested to discuss these papers in the future version.*)

In summary, the method presented in this paper essentially leverages a large pre-trained vision-language model to identify potentially correct samples and exclude likely incorrect ones. As I understand it, the method could be effectively explained within lines 253-258 alone, yet the presentation is overly complex. The authors need to restructure the manuscript to clarify the paper's contribution and explicitly compare it with relevant work.

**Questions:**

See weakness.

---

### Official Review · Reviewer_Czpz · 2024-11-04

**Soundness:** 3
**Presentation:** 3
**Contribution:** 2
**Rating:** 5
**Confidence:** 3

**Summary:**

This paper introduces a model-agnostic noise mitigation paradigm for the limitations of current noisy label approaches. It leverages cosine similarity measures to distinguish between noisy and clean samples efficiently. It shows robustness across various real-world noisy benchmarks.

**Strengths:**

- This work is well-written, from the challenges and motivation to the theoretical analysis and method design.
- This paper focuses on an extended scenario from traditional classification tasks to image-text matching task.
- The proposed method also considers the computation consumptions. The efficiency analysis shows its huge potential in practical applications.

**Weaknesses:**

- The contribution of the one-step property is weakened due to the common sense that the pre-trained model performs well in distinguishing noise samples because the noisy samples do not damage it. Training a robust model from scratch from noisy datasets is more challenging and attracts more attention.
- I suggest authors conduct more experiments on noise types and noise rates especially extreme noise rates.
- I recommend experiments performed on different scoring functions.

**Questions:**

- I recommend discussing the relationships with previous works using pretrained models e.g.[1].

[1] Fine tuning pre trained models for robustness under noisy labels. *arXiv preprint arXiv:2310.17668*.

---

### Official Review · Reviewer_NjEA · 2024-11-04

**Soundness:** 2
**Presentation:** 2
**Contribution:** 2
**Rating:** 5
**Confidence:** 3

**Summary:**

The paper addresses the challenge of mitigating the detrimental effects of noisy labels in large-scale pre-training tasks, where obtaining entirely clean data is often impractical. The authors propose a model-agnostic approach called One-step AntiNoise (OSA), which utilizes an estimator model and a scoring function to assess noise levels through single-step inference, significantly reducing computational costs. OSA leverages high-dimensional orthogonality to establish a robust boundary for separating clean and noisy samples, demonstrating enhanced training robustness, improved task transferability, and ease of deployment across various benchmarks and models. The paper provides a theoretical framework explaining the stability of the decision boundary and conducts comprehensive experiments to validate the method's effectiveness and efficiency. The authors conclude that OSA is a novel solution for noise mitigation in practical large-scale training scenarios, with code available for reproducibility.

**Strengths:**

- The paper introduces a novel, model-agnostic method called One-step AntiNoise (OSA) that addresses the issue of noisy labels in a cost-efficient manner, which is an advancement over existing noise mitigation techniques.

- It provides a theoretical framework that explains the stability and precision of the decision boundary in high-dimensional spaces, offering insights into why and how the proposed method works effectively.

- The paper backs up its claims with empirical evidences, demonstrating OSA's superiority across various benchmarks, models, and tasks, which strengthens the credibility of the proposed method.

- The paper shows that OSA introduces minimal additional training time compared to standard training methods, making it suitable for real-world applications.

- The paper demonstrates that OSA is not only effective in standard noise settings but also exhibits strong task transferability and model adaptability, making it a versatile solution applicable to a wide range of scenarios.

**Weaknesses:**

- Could the authors provide further insights into the design of the scoring function (Equation 5)? Specifically, what is the value of $\beta$ across different datasets and models, and how does the sensitivity of $\beta$ impact performance?

- Regarding Table 4, is it possible to generalize OSA for multi-class classification?

- In Figure 2, is it clear whether the estimator remains fixed or is updated during training? In other words, do the estimator and the target model share the same weights?

- Could the authors include time statistics for more methods in Table 7? Specifically, how is the time recorded? Since convergence time can vary among different methods, it is important to also consider this aspect.

**Questions:**

See **Weaknesses**.

---

### Note · Authors · 2024-12-23

I have read and agree with the venue's withdrawal policy on behalf of myself and my co-authors.